# Identifying the Main Drivers in Microbial Diversity for Cabernet Sauvignon Cultivars from Europe to South Africa: Evidence for a Cultivar-Specific Microbial Fingerprint

**DOI:** 10.3390/jof8101034

**Published:** 2022-09-29

**Authors:** Jordi Tronchoni, Mathabatha Evodia Setati, Daniela Fracassetti, Federica Valdetara, David Maghradze, Roberto Foschino, Jose Antonio Curiel, Pilar Morales, Ramon Gonzalez, Ileana Vigentini, Florian Franz Bauer

**Affiliations:** 1Faculty of Health Sciences, Valencian International University, 46002 Valencia, Spain; 2South African Grape and Wine Research Institute, Stellenbosch University, Stellenbosch 7602, South Africa; 3Department of Food, Environmental and Nutritional Sciences, Università degli Studi di Milano, 20133 Milan, Italy; 4Faculty of Viticulture and Winemaking, Caucasus International University, 0141 Tbilisi, Georgia; 5Department of Biomedical, Surgical and Dental Sciences, Università degli Studi di Milano, 20122 Milan, Italy; 6Instituto Nacional de Investigación y Tecnología Agraria y Alimentaria, 28040 Madrid, Spain; 7Instituto de Ciencias de la Vid y del Vino, CSIC, Gobierno de la Rioja, Universidad de La Rioja, 26006 Logroño, Spain

**Keywords:** meta taxonomics, mycobiome, vine cultivar, wine grape

## Abstract

Microbial diversity in vineyards and in grapes has generated significant scientific interest. From a biotechnological perspective, vineyard and grape biodiversity has been shown to impact soil, vine, and grape health and to determine the fermentation microbiome and the final character of wine. Thus, an understanding of the drivers that are responsible for the differences in vineyard and grape microbiota is required. The impact of soil and climate, as well as of viticultural practices in geographically delimited areas, have been reported. However, the limited scale makes the identification of generally applicable drivers of microbial biodiversity and of specific microbial fingerprints challenging. The comparison and meta-analysis of different datasets is furthermore complicated by differences in sampling and in methodology. Here we present data from a wide-ranging coordinated approach, using standardized sampling and data generation and analysis, involving four countries with different climates and viticultural traditions. The data confirm the existence of a grape core microbial consortium, but also provide evidence for country-specific microbiota and suggest the existence of a cultivar-specific microbial fingerprint for Cabernet Sauvignon grape. This study puts in evidence new insight of the grape microbial community in two continents and the importance of both location and cultivar for the definition of the grape microbiome.

## 1. Introduction

Phylogenetic surveys through targeted amplicon sequencing have, in the past five years, generated valuable insights into global microbial diversity, including that of vineyards (bark, fruit, leaves, and soils) and different vine cultivars. These studies show that natural microbial communities significantly impact grape and soil health, while also strongly influencing oenological processes and the organoleptic character of wine [1,2]. The data suggest that vineyard and grape microbiomes are significant contributors to “terroir”, a concept that is highly valued in the global wine industry and refers to all the elements, anthropic and natural, that provide wines with regionally distinct aromatic and taste profiles [3]. A better understanding of the factors that impact microbiome distribution and microbial biodiversity is therefore essential to better harness natural ecosystems for quality wine production.

Studies thus far have revealed the impact of both anthropic and natural environmental factors in vineyard and grape microbial biodiversity. In particular, the data show that species richness and diversity are influenced by vineyard management systems [4,5,6,7], while different ecological niches within a vineyard are characterized by diverse microbial populations [5,8,9,10]. Other data suggest that vineyard location as well as grape varietal might be strong determinants of microbial community composition [11,12,13,14]. However, the extent to which these factors might shape microbial community structures could vary depending on the spatial distances considered [9,15]. For instance, differences between cultivar microbiome compositions might be more pronounced at a local level than over larger scales [9,13,14]. Varietal features such as grape bunch compactness as well as berry skin thickness have been highlighted as factors that could contribute to cultivar differences [13], while microclimatic conditions and vineyard geographical orientation, as well as geographical coordinates, were highlighted as important drivers of regional distinctions in grape microbial community structures [11,14].

A microbiomics approach allows us to investigate and evaluate bacterial and fungal diversity of different samples. Apart from a huge study on the global topsoil microbiome [13], examples in wine science have been focused on the microbial diversity of vineyards (bulk, fruit, leaves, and soil) and cultivars.

Bahram et al. (2018) [13] evaluated the microbiome of vineyard topsoil and showed that the genetic diversity among bacterial species is highest in temperate habitats and the geographic distance impacts less strongly than environmental variables on the microbial gene composition. Indeed, they also demonstrated that fungi and bacteria show a global niche differentiation directly correlated with external factors (i.e., rainfall and soil pH).

Different single-cultivar studies analyzed intra-varietal microbial differences within smaller [6,8,10,16] or larger [4,5,9,15,17,18] areas, focusing on bacterial [8,10] or fungal [6,9,15,16,17,18,19] composition, or analyzing both groups of microorganisms [4,5]. In particular, some works estimated the differences in microbial biodiversity linked to vineyard management and spatial or biological niches (bulk, fruit, leaves, and soil) [5,8,9,10,15,17,18,19]. The results demonstrated that species richness and biodiversity are influenced by viticultural practices [4,5,6,16], and operational taxonomic units (OTU) peculiar to specific farming system were described [6,19]. Moreover, different niches are characterized by a diverse microbial population [5,8,9,10]. The data suggest that dissimilarity in fungal community composition increases with spatial distance. Several studies have highlighted regional differences in fungal communities: Taylor et al. (2014) [15] reported that the species richness of grape-associated fungal communities (species counts) shows regional differentiation in New Zealand, while species abundances is relatively stable. Data by Liu et al. [17] suggest regional patterns in Pinot Noir grape musts in six regions in Southern Australia. In particular, *Saccharomyces cerevisiae* and grapevine-associated species (such as *Aureobasidium*) appeared in different abundances, with distances between vineyards ranging from 5 km to 400 km. The data suggest that weather and soil properties affect the must fungal diversity, confirming what has been reported in other large-scale studies where mainly precipitation and C/N ratios could be considered predictors of soil fungal richness and community composition. Finally, Li et al. [18] reported distribution patterns of fungal communities at the regional level in Chinese wines from Marselan grape. The results showed that some fungi could be considered regional biomarkers because their relative abundance changed across six regions. These fungal biomarkers included dominant (i.e., *Alternaria*, *Aureobasidium*, *Rhodotorula*, and *Cladosporium*), low abundant (i.e., *Aspergillus*, *Papiliotrema*, and *Phoma*), and phytopathogenic (i.e., *Colletotrichum*, *Botrytis*, and *Aspergillus*) genera.

However, to date, few studies based on sequencing techniques have directly compared grape microbial communities associated with different cultivars at small (<100 km) and large (>100 km) geographical scales [11,12,13,14,20]. Recently, in order to provide evidence for microbial contributions to wine terroir, Liu et al. [20] analyzed the fungal distribution associated with Pinot Noir and Chardonnay vineyards located within a 12 km radius in Australia. Although the two varieties were characterized by a specific fungal signature in terms of genus abundance, a set of ubiquitous fungi belonging to *Aureobasidium*, *Cladosporium*, *Saccharomyces*, and *Rhodotorula* genera were identified in both cultivars. The study revealed that geographical origin had a greater impact on the fungal community than grape variety, with Pinot Noir showing a stronger geographical differentiation than Chardonnay. On the other hands, studies carried out considering larger distances (>100 km) revealed that different wine-producing regions possess distinct, distinguishable fungal microbiota. Bokulich et al. (2014) [11] investigated fungal and bacterial communities in Chardonnay, Cabernet Sauvignon, and Zinfandel must samples across California. As for the grape-associated fungi, *Capnodiales* (including *Cladosporium* spp.) and *Penicillium* were significantly more abundant in Chardonnay; *Dothideomycetes*, *Agaricomycetes*, *Tremellomycetes*, *Microbotryomycetes*, and *Saccharomycetaceae* in Cabernet Sauvignon; and *Eurotiomycetes* (*Aspergillus*), *Leotiomycetes*, and *Saccharomycetes* (mainly *Starmerella bacillaris* ex. *Candida zemplinina*) in Zinfandel. While the fungal taxonomic dissimilarity remained highly stable over two years, a wide variation in the index was found in individual wine-growing regions, demonstrating that the grape variety played a significant role in shaping the biogeography of microbial communities. However, the study relied on the sampling of musts within wineries, and other elements such as harvesting practices will have significantly influenced the data.

Bacterial and fungal communities appeared separated comparing two completely different fruit species, apple and blackcurrant, in two geographical regions. Within the same fruit samples, the blackcurrant-associated microbial population resulted as greatly different, considering the geographical origin, while in apple samples the dissimilarity was greater among fungal populations than the bacterial ones [19].

The purpose of this work was to evaluate the variability of grape microbiota on a global scale in differently distributed varietals: a varietal that has been globally planted, Cabernet Sauvignon, and several varietals whose distribution is restricted, or mostly restricted, to one specific country or region. Four different areas of the world were chosen for this analysis. To further investigate the local variance, for each country an average of four sampling sites for each variety were analyzed. To the best of our knowledge, this is the first work with an “international vs. local approach” that explores the variability of distant geographic places contemporarily and deals with them by decreasing the differences in the experimental approach, which would sometime limit the possibility of comparisons between such datasets.

## 2. Materials and Methods

### 2.1. Grape Sampling and DNA Extraction

Samples were collected at four different regions: three from Europe and one from South Africa. The wine regions were La Rioja in Spain, Tuscany in Italy, Kakheti in Georgia, and Stellenbosch in South Africa. In each region, different localizations were selected for sampling, five in the case of La Rioja, four from Italy and Georgia, and three from South Africa. From each localization, samples were collected from Cabernet Sauvignon grapes and from a second, locally prominent or unique cultivar, referred to as local cultivar (or others). The local cultivars were Tempranillo in La Rioja, Sangiovese in Tuscany, Rkatsiteli in Kakheti, and Chenin Blanc in Stellenbosch (Table A1). The two prerequisites for the selection of location were: (1) vineyards of both cultivars in the same region had to be in close spatial proximity, ideally adjacent, thus sharing climatic conditions; (2) they had to be subjected to similar agricultural and agronomic practices. In particular, the wine estates included in the study carried out an integrated pest management (IPM) farming method for the rational control of harmful organisms for plants. Differences in the number of sample localization are due to the difficulties in finding vineyards fulfilling these conditions. From each vineyard, clusters were harvested from random plants, in equal numbers, from the sunny and shaded sides of the row. The entire vineyard area was covered, and a total amount of 10 kg of grapes was collected. Undamaged samples, healthy bunches from both sides of the panel (i.e., bunches that receive morning and afternoon sun) were collected. In the lab, 5 kg of grapes were hand destemmed, crushed in a sterile bag, and the resulting juice homogenized. Fifty milliliters of juice sample was centrifuged at 8500× *g* for 10 min. The pellet was washed three times with a 50 mL EDTA-PVP solution (0.15 M NaCl, 0.1 M EDTA, 2% (*w*/*v*) Polyvinylpyrolidone), followed by 3 washes with 50 mL TE buffer (10 mM Tris-HCl pH 7.5, 1 mM EDTA pH 8). After these steps, the pellet was frozen at −80 °C until DNA extraction. Genomic DNA was extracted from approximately 500 mg of pellets following the soil DNA extraction kit instructions (PowerSoil DNA Isolation Kit, Qiagen, Hilden, Germany; Mo Bio and SurePrep Soil DNA Isolation Kit, Fisher Scientific, Waltham, MA, USA).

### 2.2. Sequencing Library Construction

Amplification of the ITS1-5.8S rDNA-ITS2 was performed using fusion primers based on the BITS/B58S3 primer pair designed by Bokulich and Mills (2013) [21] and Nextera platform-specific adaptor sequences, according to Setati et al. (2015) [6]. The PCR was performed in 25 μL reactions containing 1 × Ex-Taq buffer, 0.2 mM dTNPs, 0.25 μM of each primer, and 100 ng DNA template (with a 260/280 ratio ≥ 1.8). Triplicate reactions were performed for each DNA sample. Cycling conditions consisted of an initial denaturation at 94 °C for 3 min, followed by 40 cycles of denaturation at 94 °C for 30 s, annealing at 55 °C for 30 s, and extension at 72 °C for 45 s, and a final extension of 10 min at 72 °C. The PCR products were purified using the Zymoclean Gel DNA recovery kit (Zymo Research, Inqaba Biotechnical Industries, Pty Ltd., Pretoria, South Africa) and quantified using the NanoDrop 1000 spectrophotometer (Thermo Scientific, Waltham, MA, USA). The amplicons from triplicate PCR reactions were combined at equal concentrations and used for Nextera library preparation and sequencing. Samples were subjected to standard quality control measures (fluorometric quantification and normalization). One nanogram of each amplicon pool was used in a standard indexing PCR protocol for a paired-end sequencing library (using Nextera XT DNA Library Prep workflow), and samples were sequenced using MiSeqV3 chemistry (2 × 300 reads) (Illumina, San Diego, CA, USA).

### 2.3. Data Analysis

Raw Illumina fastq files were subjected to quality analysis using FastQC, and low-quality sequences with a Phred score below 30 were identified using dynamic trimming and removed. Different QIIME version 1 scripts were used [22]; except when indicated between brackets, standard parameters were used: validate_mapping_file.py to validate data format; split_libraries_fastq.py (--barcode_type ‘not-barcoded’) for multiplexing; truncate_reverse_primer.py to remove reverse primer and subsequent sequences (--barcode_type ‘not-barcoded’); identify_chimeric_seqs.py to remove chimeras with the identifying algorithm usearch61 (--m, --chimera_detection_method usearch61, --suppress_usearch61_ref); pick_open_reference_otus.py for OTU picking steps (--suppress_align_and_tree, OTU clustering threshold is 97%); and summarize_taxa.py to create taxa summary tables. OTU and taxonomy tables with metadata files were used to feed MicrobiomeAnalyst web-based software [23]. Unless specified, filter parameters were a low count filter of 2, with a prevalence in sample of 10%. Data were normalized through the Total Sum Scaling (TSS) method. Different R packages were used: EdgeR [24] and DESeq2 [25] for statistical comparison of differential OTUs in samples and conditions; Vegan [26] for rarefaction curve analysis. To investigate richness, the main alpha diversity indexes were used (Shannon and Simpson, Chao1). The Bray–Curtis dissimilarity index and Permutational Multivariate Analysis of Variance (PERMANOVA) were used to compare fungal community composition between regions [20], and the beta diversity was visualized using the ordination-based method of Principal Coordinates Analysis (PCoA). ANOVA tests were used to identify significant differences in both analyses. In Hierarchical Clustering (HCl) and Heatmap visualization, the Ward clustering algorithm was used.

## 3. Results

### 3.1. Design of Experiments and Sequence Analysis

To understand the main drivers in microbial diversity in wine grapes from different vine cultivars, we sampled different regions from Europe (Georgia, Italy, Spain) and South Africa (Table A1). From each location, grapes were collected from an international cultivar, Cabernet Sauvignon, and from a second, locally prominent or unique cultivar. Illumina paired end sequencing of the genomic DNA extracted from grape musts was used to explore the fungal biota (mycobiome) of the different vineyard samples, and ITS1-5.8S rDNA-ITS2 libraries were generated. A dataset containing forward reads (mainly containing partial ITS1-5.8S sequences) was selected for further analysis; indeed, this last proved to be more informative than the joined reads (ITS1-5.8S-ITS2) that significantly reduced the amount of available data. Sampling depth and sequencing coverage were analyzed by rarefaction curves, showing that in most cases the desired plateau was reached, especially for samples coming from the Italian and Georgian datasets (Figure A1). The analysis of the sample diversity, using the main alpha diversity indexes, showed that the South African and Spanish datasets were the most diverse, with the former having the least variance between samples (Figure 1).

### 3.2. Fungal Distribution

A total of 361,981 ITS high-quality sequences were generated from 32 grape samples, and they were clustered into 92 fungal OTUs with 97% pairwise identity (Assigment between OTU numbers and genera is shown in Table A2). Most of the species belonged to the Ascomycota phylum (Figure A2a). Basidiomycota species represented an important percentage (25%) in only a few samples across the different datasets; for instance, one sample in the Georgian and one in the South African dataset, with the lowest representation in the Italian samples. Some Spanish samples (in both varietals) and one Cabernet Sauvignon sample in South Africa showed incidence of the Mucoromycota phylum (within 8% as a maximum). Dothideomycetes were the most common fungi across all samples, with a high presence of Leotiomycetes in the Italian samples, together with Eurotiomycetes in the Spanish ones. The presence of Saccharomycetes was also important in some isolated samples from the Italian (showing the highest values), South African, and Spanish datasets (Figure A2b).

Genera abundance distribution can be seen in Figure 2. As expected, some yeast genera had an important presence across all datasets, as is the case for Aureobasidium. Although only healthy samples were collected, the second fungal genus in abundance was Botrytis, which, however, showed very high heterogeneity in its quantitative distribution.

Despite these heterogeneities, some patterns can be observed: South African samples showed an important presence of Alternaria in all samples, with significantly higher frequencies when compared to all the other samples; an important abundance of Penicillium was found in the Spanish samples, and Georgian samples contained high frequencies of unidentified fungi.

### 3.3. Deconstructing the Drivers Defining Microbial Population

#### 3.3.1. Country of Isolation Is the Primary Driver

A hierarchical clustering was made to show similarities among groups of samples (Figure 3). The results highlighted that all the South African samples separated from the rest of the European samples in a single sub-cluster, which, however, also included two Spanish samples. The second cluster contained all the European samples, with the remaining Spanish samples forming their own sub-cluster separated from the Italo-Georgian sub-cluster. Even so, in this last group the samples belonging to the same country tended to be closely associated. The figure also shows how the vine cultivar from which the samples were isolated seems to have little effect in defining the groups; the samples of the local cultivars grouped together only in some cases. When they did, it was mostly true for two adjacent samples, as was the case in samples of Chenin Blanc in South Africa or of Sangiovese and Rkatsiteli in the Italo-Georgian subgroup. The international cultivar samples showed a similar arrangement, except in the case of Georgia, where the largest grouping of Cabernet Sauvignon samples is observed.

#### 3.3.2. Cabernet Sauvignon Vine Cultivar Shows a Specific Microbial Fingerprint

Despite country of isolation being the main driver of variation, other factors were influencing the outcome (Figure 4).

When comparing datasets from Cabernet Sauvignon grapes against local cultivars by PCoA, the main component explaining the distances between samples is the principal component in the local and Cabernet Sauvignon cultivars, explaining 39.1 and 27.1% of differences each (Figure 4a,b). The closer samples are to each other, the more tightly they cluster in a final plot. As it can be seen in the PCoAs, Cabernet Sauvignon samples do not appear to be driven by axis one or two, forming a cloud of points altogether. On the contrary, the local cultivars appear to be more clearly stratified along the PC2 axis: from top to bottom, Georgia, Italy, Spain, and finally South Africa, with the latter being the most closely grouped.

A similar situation can be observed with HCl analysis of the two datasets (Figure A3). Local varieties group very similarly to the previously described HCl (when all samples, international and local, were analyzed together) (Figure A3a). The South African samples formed an independent group from the European samples, this time without the inclusion of any Spanish sample (Figure A3b). Differences between Cabernet Sauvignon and local cultivars are not related with the most represented genera (Figure 5). The most represented genera, *Aureobasidium* and *Botrytis*, had similar percentages in both groups and roughly constituted half of the fungal species observed. On the other hand, the fungi that constituted the other half of the diversity varied more significantly between both groups. This was especially true for the species that appeared with lower abundances, which showed very different percentages between cultivars. The average diversity of the Cabernet Sauvignon cultivars was greater than that of the local cultivars (Figure A4). This was also observed in the “Other” category of Figure 5, which groups species with a low incidence; this category had greater weight in Cabernet Sauvignon compared with local cultivars. From the OTUs shared by both cultivars, five were significantly different between grape cultivars (Figure A5); these corresponded to *Rhodotorula*, *Cystobasidium*, *Papiliotrema*, *Filobasidium*, and the “Other” category.

#### 3.3.3. Drivers of Differentiation at a Local Level

We also analyzed samples from each country individually (Figure A6, Figure A7, Figure A8 and Figure A9). In general, in each country of isolation, samples of Cabernet Sauvignon and the corresponding local cultivar coming from the same field clustered together. The number of vineyards showing this behavior differed from region to region; Georgia and Italy had the higher percentages with 75% (3 out of 4 vineyards) each, Spain 40% (2 out of 5 vineyards), and South Africa 33% (1 out of 3 vineyards). This clustering was analyzed for differences in OTUs, with only Italy, Spain, and South Africa presenting OTUs that were significantly different among fields (Figure 6). In the case of South Africa, with only vineyard number 3 clustering Cabernet Sauvignon and Chenin Blanc together, the differential abundance analysis showed that the only significant OTU comparing the three vineyards was OTU34, which corresponds to Botrytis, which was also in higher abundance in vineyard 3. Two OTUs appeared significantly different in the Spanish samples; one corresponded to the category “unidentified” (OTU61) and the other to Rhizopus (OTU90). The unidentified category collected the reads that did not have a blast match and indicated that, in the case of vineyard 1, this category was more abundant. For Rhizopus, three out of the five vineyards presented this fungus. Both OTUs have none or very low representation in fields 4 and 5, which are the ones clustering in the HCl. In Italy, where fields 1, 3, and 4 had samples from Cabernet Sauvignon and Sangiovese next to each other, the only significant OTU corresponded to Neoscytalidium (OTU5), which was higher in the last two fields. No significant OTUs appeared in the Georgian samples.

#### 3.3.4. Yeast Genera May Shape Population Diversity

In order to evaluate how the presence of different genera was driving the grape mycobiome, a correlation analysis was performed (Figure 7). In the correlation matrix, it can be observed that OTU34 had a strong negative correlation with several other species. This OTU corresponded to *Botrytis* and showed the strongest negative correlation of the entire matrix, especially on *Aureobasidium* (OTU12). In general terms, *Aureobasidium* was the most abundant OTU revealed in this study, except when *Botrytis* was present. On the other hand, several OTUs had an important positive correlation over other OTUs, for instance, *Bipolaris*, *Exserohilum*, and *Cystobasidium*. In *Bipolaris* and *Exserohilum* the effect is reciprocal and on unidentified species, in *Cystobasidium* the effect is on unidentified species. Contrary to what happens with *Aureobasidium* and *Botrytis*, which represent two of the major groups observed in the population, *Bipolaris*, *Exserohilum*, and *Cystobasidium* constitute much lower percentages.

## 4. Discussion

In this work, we can distinguish up to four layers of influence in the distribution of the microbiota.

Country of origin, the region of isolation of the samples, is the most influential layer. This evidence confirms what has been reported in other studies, where both single- and multiple-cultivar analysis highlighted that fungal microbiome correlates with regional grape microbial patterns [5,8,9,10,17,18,20]. Since the initial treatment of the data, datasets coming from the different localizations show specific characteristics in the form of different abundances, richness, or number of unidentified fungi. These noticeable differences, easily seen in the HCl (Figure 3), group the European samples on one side, separate from the South African samples. The presence of *Alternaria* in the South African dataset is higher compared to the other datasets and consistent through the samples. This result has been recently confirmed in fungal profiles obtained from Cabernet Sauvignon grapes collected from a biodynamic vineyard in the Stellenbosch wine producing region of South Africa [16]. The filamentous fungus *Alternaria* sp. is known as a pathogenic grapevine endophyte that can became more abundant along specific developmental stages of grape; for example, *Alternaria infectoria* and *Alternaria rosae* were only detected in grape samples at veraison and appeared in high occupancy afterwards [27]. *Alternaria* is a plant pathogenic fungus that produces a mycotoxin dangerous to human health [28], which can reach the wine and whose presence has recently been suggested to have increased due to climate change [29]. Moreover, the analysis of the fungal microbiota from Pinot Noir and Chardonnay vineyards from different cellars (located at a small-scale pairwise distance of 8–12 km) displayed that *Alternaria* sp. Was detected only in samples from one wine estate, suggesting that not only local climatic conditions significantly correlate with microbial compositions in grape musts [11], but also microclimatic changes on field blocks could play a role in shaping the fungal community diversity.

A second layer of differentiation whichs the wine cultivar fromwhichch samples where isolated. Although the influence of the cultivar has been previously described [11], this is the first time that influence is observed in cultivars located in different countries; although the present work analyzed the mycobiota in two continents, this finding provides evidence that the grape variety could also act as a factor affecting the composition of fungal communities at a global scale. Further metagenomic investigations should include retrieving grape from other regions located on different continents, to compare them with those here analyzed to confirm our previous hypothesis. Moreover, the analysis of the data does not allow us to establish exactly what are, in terms of species and abundances, the characteristics that this fingerprint possesses for the Cabernet Sauvignon cultivar, although the results indicate that the less abundant species could define this fingerprint, requiring deeper sequencing for identification.

The third detected layer of influence is the sampling field, because samples from the same field, either the common or the local cultivars, are in some cases grouped. It must be considered that, in this work, an effort was made so that the local effects were as homogeneous as possible between fields in the collection of the samples from common and local cultivars. This result may be due to different causes, most likely human practices related to the management of the vineyard and the different agricultural techniques that may be influencing the observed grouping. Similarities in the fungal communities among different wineries in the same regions has been recently reported [20]; however, thought our output confirms this evidence, further studies are required to establish potential factors driving the fungal community profiles on vineyard blocks (or fields) with different microclimatic, viticultural, and topographical conditions, beyond the aim of the present work. In particular, at the local level, the genera that were significantly different between fields were *Botrytis* in South Africa, *Rhizopus* in Spain, and *Neoscytalidium* in Italy. In this case, the climatic conditions could have determined the success in the development of some pathogenic fungi, but further analyzes are necessary to verify a possible correlation between the climatic parameters and pathogenic fungi found locally, considering both the vintage under study and the three interested wine-producing areas. *Botrytis* is a genus of anamorphic fungi in the family *Sclerotiniaceae*, which includes *B. cinerea*, an airborne filamentous fungus that causes grey mold disease [30]. This necrotrophic phytopathogenic fungus is ubiquitous, and climate conditions including mean temperature, humidity, and precipitation are considered the main variables influencing the botrytization process [31,32]. *Rhizopus* was found to be the main components of the wine grape mycobiota in Tokaj grape berries of Slovak regions and part of the fungal community in Slovakian wine grapes from small Carpathians wine region [33,34,35]. Some species of this *genus* provoke *Rhizopus* rot, common on soft fruits, more abundant in warm, humid climates than in cool climate viticulture. Finally, the *genus Neoscytalidium*, belonging to the *Botryosphaeriaceae* family and includes species associated with *Botryosphaeria* dieback on grapevines. In addition, for this taxon, the geographical distribution of some *Botryosphaeriaceae* species has been shown to be associated with climate [36].

The data also identify another influencing factor, “altering species”, usually grape pathogens, whose presence acts by displacing and varying the fungal and yeast population. In some cases, their presence favors the appearance of other genera, such as the positive effect of the presence of *Botrytis* on the appearance of *Diplodia* from the family *Botryosphaeriaceae*, which is associated with *Botryosphaeria* canker and other trunk diseases of grapevine [37]. However, most frequently this interaction can be negative [38]; thus, the negative correlation observed in our samples between *Botrytis* and *Aureobasidium* would deserve an in-depth study in order to assess whether it is explained by biological interactions between the two species. Although *Alternaria* is also a pathogen, more highly represented in the South African samples when compared to the rest of the samples, we cannot identify its effect on other species of microorganisms in our dataset, as similar frequencies of this fungus appear in all samples.

The study also lacks microbial reference data for less-studied wine regions, as is the case of samples collected in Georgia, which have the highest rate of unidentified OTUs. This finding further confirms that, despite the numerous efforts of identification of microorganisms from the vineyard ecosystem, there are locations with microbiological particularities that should be explored in greater depth. Although such unidentified species are generally found in relatively low abundances, they may well exert a significant ecological impact. In most microbiomic datasets, species (OTUs) of low abundance are those that present the greatest diversity, but also have the least available information regarding their role. Whether or not this abundance is sufficient to constitute a biological impact is another matter. It is generally accepted that microbiological diversity constitutes a good indicator of vineyard health and can act as a buffer against harmful invasive species by increasing the likelihood of having local antagonist fungi present.

This meta-analysis of grape microbiomes represents the first study to include two continents, within a more global vision. Through standardization of sampling design, sample preparation, and metagenomic data generation, the study provides an opportunity to evaluate several relevant issues with regards to microbial diversity in vineyards, and in particular contributes significant confirmation from existing hypotheses and new insights regarding the relative importance of both location and cultivar in defining grape microbiome.

The data indeed suggest that, on a global scale, location has a significant impact on microbial diversity, whether with regards to unique genera or species, or with regards to species distribution. They also confirm the presence of a common core of genera that is present globally in vineyards and on grapes, suggesting that vineyards would provide an ideal model to evaluate the evolutionary trajectory of anthropogenic microbial ecosystems.

The results also suggest that grape varieties are indeed characterized by unique microbial fingerprints, because the Cabernet Sauvignon samples from all locations showed closer association than the samples from country-specific varietals. However, a signature in terms of species isolation or species distribution cannot be defined in this study, and such an analysis will require additional datasets and deeper sequencing than was the case here.

## Figures and Tables

**Figure 1 jof-08-01034-f001:**
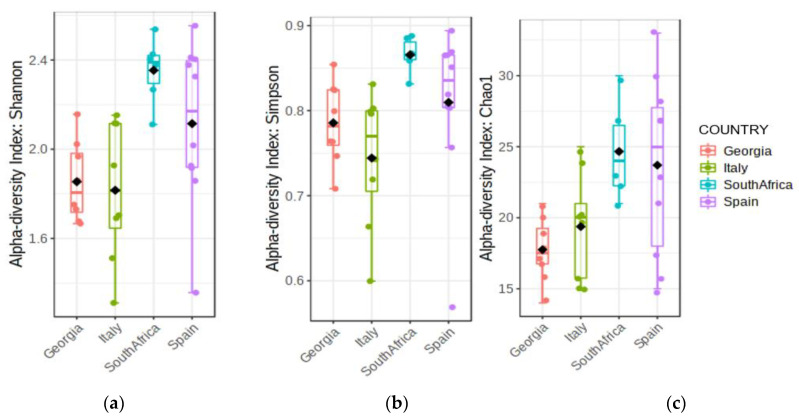
Analysis of the sample diversity: alpha-diversity index: (**a**) Shannon, *p*-value: 0.004172; (ANOVA) F-value: 5.5182; (**b**) Simpson, *p*-value: 0.02898; (ANOVA) F-value: 3.4793; (**c**) Chao, *p*-value: 0.012439; (ANOVA) F-value: 4.339. The colored solid line inside the box is the median value, and the black rhombus is the mean value; the colored dots are individual samples.

**Figure 2 jof-08-01034-f002:**
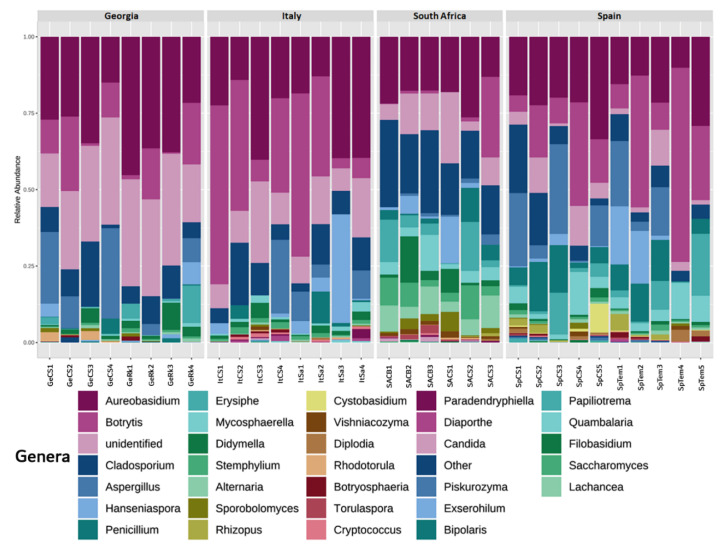
Relative abundance and taxonomic assignment of genera in the 32 samples from the four countries. Sample code: Ge, Georgia; It, Italy; SA, South Africa; Sp, Spain; CS, Cabernet Sauvignon; Rk, Rkatsiteli (Ge); Sa, Sangiovese (It); Tem, Tempranillo (Sp); CB, Chenin Blanc (SA).

**Figure 3 jof-08-01034-f003:**
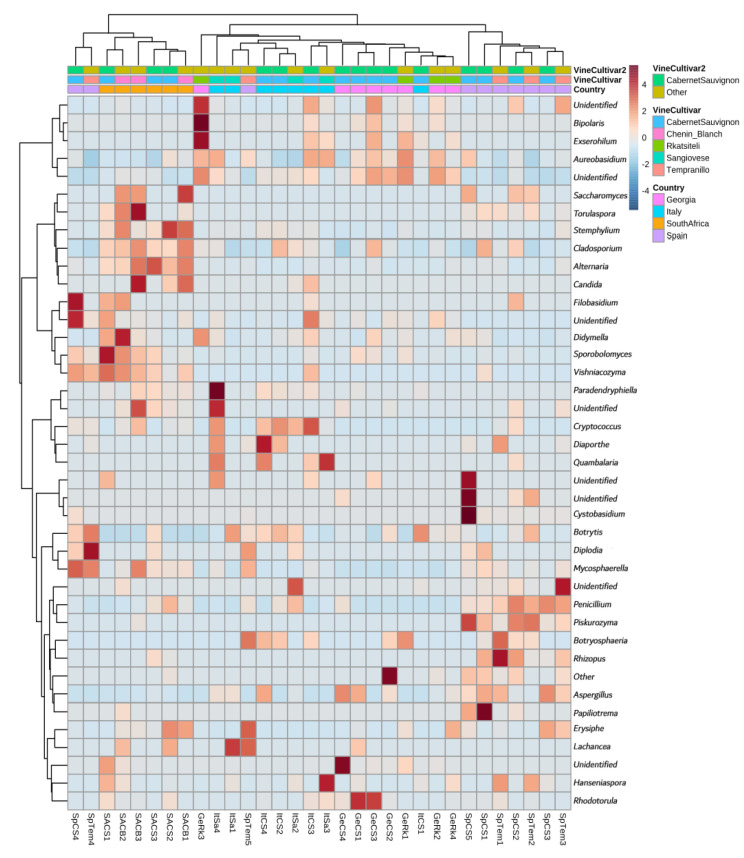
Clustered heatmap showing the variation of genus abundance with regard to vine cultivars and countries. Sample code: Ge, Georgia; It, Italy; SA, South Africa; Sp, Spain; CS, Cabernet Sauvignon; Rk, Rkatsiteli (Ge); Sa, Sangiovese (It); Tem, Tempranillo (Sp); CB, Chenin Blanc (SA).

**Figure 4 jof-08-01034-f004:**
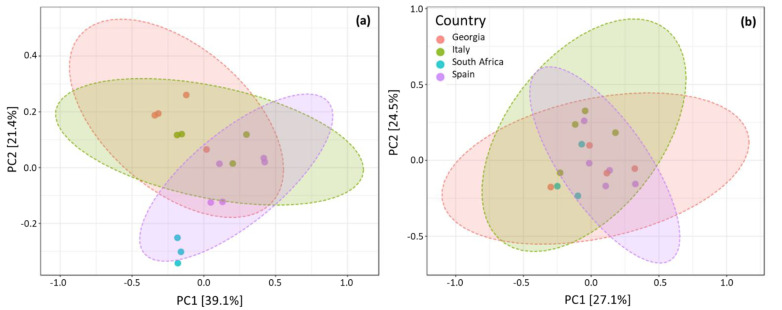
Two-dimensional PCoA for Local (**a**): (PERMANOVA) F−value: 1.925; R−squared: 0.3249; *p*−value: 0.013) and Cabernet Sauvignon (**b**): (PERMANOVA) F−value: 3.657; R−squared: 0.4776; *p*−value: 0.001) of the Georgian (red), Italian (green), Spanish (light blue), and South African (blue) samples.

**Figure 5 jof-08-01034-f005:**
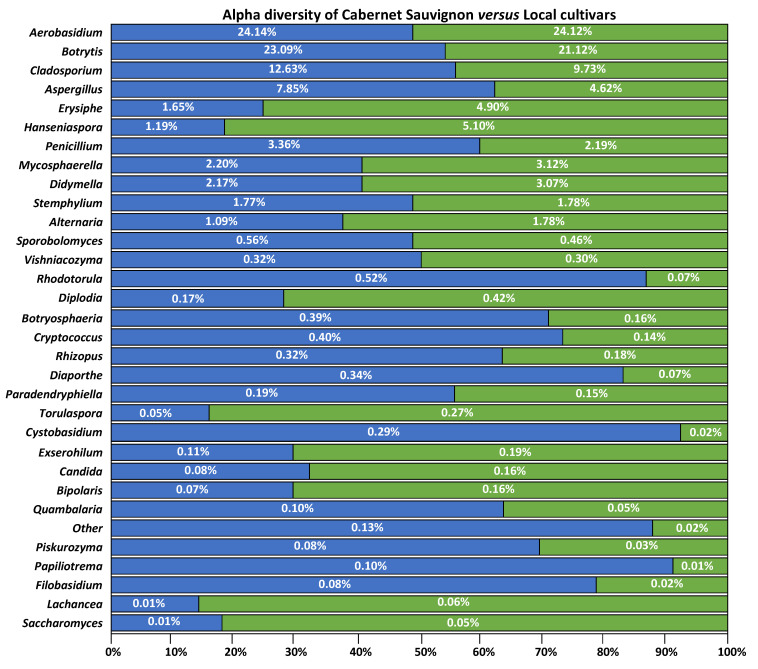
Abundance percentage of each genus among Cabernet Sauvignon cultivars (blue bars) and local cultivars (green bars). In white, the percentage of abundance of that genus for the Cabernet Sauvignon cultivar and local cultivars.

**Figure 6 jof-08-01034-f006:**
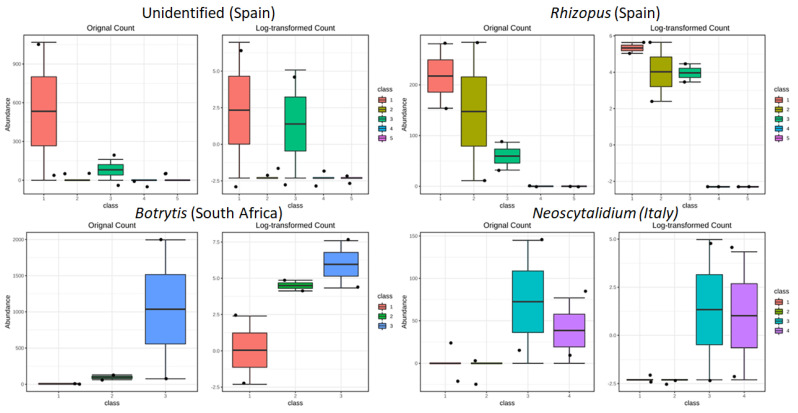
Statistically different *genus* abundance among fields: Spain, *Rhizopus* and an unidentified *genus*; South Africa, *Botrytis*; Italy, *Neoscytalidium*. Class, on the *x*−axis, indicates the field where the samples were collected (Table A1). Data analysis was performed with R package DESeq2. Statistically differentiated genus abundance corresponds to a q−value cutoff of 0.05.

**Figure 7 jof-08-01034-f007:**
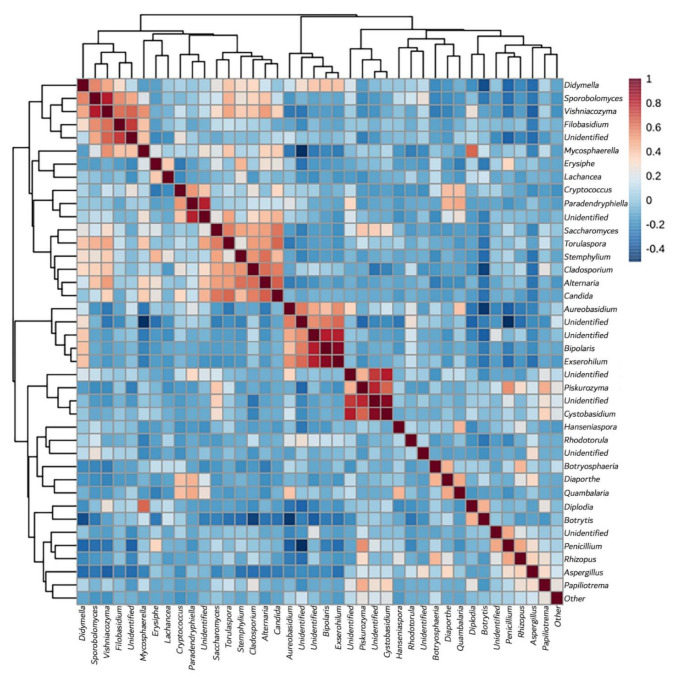
Correlation analysis for OTU abundances using Pearson’s correlation coefficients.

## Data Availability

The dataset supporting the results of this article is available in the NCBI repository under BioProject PRJNA860570. The dataset supporting the results of this article is included in the article (and its Additional files).

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
