# Peer review of "Identifying the Main Drivers in Microbial Diversity for Cabernet Sauvignon Cultivars from Europe to South Africa: Evidence for a Cultivar-Specific Microbial Fingerprint"

_jof, 2022, doi:10.3390/jof8101034_

Round 1
Reviewer 1 Report
The manuscript needs to be rewritten and some analyses should be added, especially the statistical analyse. The authors evaluated the microbiota associated with grapes with a wide-ranging coordinated approach. The influence of location and cultivar is given, revealing country-specific differences, and the negative impact of Botrytis cinerea on other fungal species.
(1) The Introduction the authors mentioned too many details on bacterial community instead of fungal community, weakening the insights gained. Some sentences are hard to understand, like Line 87-88.
(2) The version of the software and the thresholds (e.g., OTU clustering threshold) should be mentioned to improve repeatability.
(3) The results part needs to be rewritten, which should be clear and concise instead of repeating the method parts.
(4) Three alpha diversity indexes are analyzed and only two are shown in the results.
(5) The figure captions are not clear, for example, I could not understand the meaning of Figure 4 a, and b. So I could not read the part meaning these figures. What is the meaning of “class” in Figure 5.
(6) The manuscript has too many small mistakes, like Line 60.
(7) Why do the authors only discuss OTU34?
(8) In the whole text, I couldn’t see any statistical analysis. It makes no sense without statistical analysis. For example, PERMANOVA analysis is a good choice to test the effect of country of origin.
(9) Too few references in the discussion part.
(10) I wonder whether two continents could represent a global scope.
Author Response
Dear Reviewer 1,
in the attached file you find the answers to your comments. Thank in advance for the revision of our manuscript and your time.
Best regards,
Ileana Vigentini

Reviewer 2 Report
The manuscript attempts to describe the fungal diversity in vineyards located in diverse parts of the world, for which presumably similar agricultural practices and sampling strategies have been used.
As the main concern and shortcoming of the manuscript, I highlight the absence of accurate and detailed statistical interpretation of the data. The main conclusions of this manuscript are not supported by the results shown. The statistical analysis is inadequate and the presentation of the data misses critical information. Lastly, the title does not accurately capture the scope and results of the project.
Specific comments:
1. Sampling is an important parameter of this study. Please describe the locations (geographical coordinates) for all samples collected. Ideally, map(s) should be provided in the supplemental section to show the approximate location of the local cultivars and CS vineyards to support the authors' assertions.
2. In 2.3 Data analysis, Chao is mentioned as being used, however, the Results only show Simpson and Shannon diversity.
3. Please add details of the agricultural and agronomical practices used in the vineyards sampled.
4. In Figure 1, statistical parameters, and significance (if any) are not shown (p-value, F-value)
5. Figure 2: increase font size on the X-axis
6. Figure 3 is challenging to comprehend; the legend is incomplete: add details on the taxonomic level used for analysis; I suggest replacing OTU numbers with taxa names to streamline with the presentation in the main text.
7. Figure 4: the main text here does not accurately capture the results shown in the figure. More importantly, the statistical interpretation of the results the lacks completely. In Fig 4A, PC1 appears as the main driver of the diversity, yet the authors focus on PC2 (~ 20% of the variation). In Fig 4B, contrary to what the figure shows, the authors conclude that CS cultivars cluster more closely. There are no statistical parameters shown in the figure. Overall, the figure and associated text appear disjointed and the conclusions are erroneous.
8. In Table 1 what are the authors referring to when stating that “the fungi that constituted the other half of the diversity varied more significantly between groups”; give concrete examples with percentages. The authors should also note that for low-abundance taxa, the error rate is higher compared to highly abundant taxa.
9. Figure 5: please add statistical parameters to show significance. Use taxa names in addition to OTU nr.
10. In Fig 6, again, using either OTU nr or names is confusing for the reader; where in the figure are shown Bipolaris, Cystobasidium, and others mentioned in the main text?
Author Response
Dear Reviewer 2,
in the attached file you find the answers to your comments. Thank in advance for the revision of our manuscript and your time.
Best regards,
Ileana Vigentini

Reviewer 3 Report
The content of the manuscript is in keeping with the theme of the journal. The content of the study shows the geographical variation in the microbial communities of the viticultural soils. I suggest that authors need to focus on the following points:
1. Authors should describe indicators such as grape yields in each region.
2. I suggest that authors do not use descriptions from the first global-scale study.
3. the relationship between environmental factors (e.g. soil physical and chemical properties, temperature, humidity, light, etc.) and microbial communities is not explored in detail in the manuscript. I would therefore suggest that the authors describe them at least in the discussion.
4. I suggest that some environmental similarities between geographical areas are discussed, which may also partially explain the similarity of microbial communities across geographical areas.
Author Response
Dear Reviewer 3,
in the attached file you find the answers to your comments. Thank in advance for the revision of our manuscript and your time.
Best regards,
Ileana Vigentini

Round 2
Reviewer 1 Report
The authors reported the microbial diversity of the grapes with a wide-ranging coordinated approach. By sequence and bioinformatics analysis, the influence of location and cultivar is stressed, revealing the grape core and country-specific microbiota. Overall, this is a well-written manuscript. However, before it could be accepted, some details should be revised.
Specific comments
(1) Line 40-133: The introduction is too long, with too many small issues mentioned. In addition, the importance of investigating microbial community in vineyards and on grapes wasn’t mentioned.
(2) Part 2.3: The software might be out-of-date (e.g., QIIME1), please consider update the version, like QIIME2.
(3) Please present the result in a concise way. For example, in Figure 1, you can add the p-value in the Figure and make it like “p < 0.05”.
(4) The manuscript still has small mistakes, like Line 223-225, “361,981” instead of “361981” and “OTU numbers” instead of “OUT numbers”. Table A2, please change “OUT” to “OTU.”
(5) Figure 5 is strange, please consider STAMP analysis. a
(6) The tables are not in standard form (three-line table) and some details should be added, like “Refer to Table A1 (continued)”
(7) What is meaning of black rhombus in Figure A4.
(8) I still wonder whether two continents could represent a global scope, especially in the Discussion part.
(9) The importance of the investigation into the microbial community of grape wasn’t stressed in this study.
Author Response
Dear Editor,
we thank Reviewer 1 and the Staff for suggestions and support in revising the manuscript.
Best regards,
Ileana Vigentini

Reviewer 2 Report
The authors adequately answered my comments, and the revised manuscript is suitable for publication
Author Response
Dear Editor,
we thank Reviewer 2 and the Staff for suggestions and support in revising the manuscript.
Best regards,
Ileana Vigentini
Reviewer 3 Report
The manuscript has been revised and has improved considerably, and I recommend that it be accepted.
Author Response
Dear Editor,
we thank Reviewer 3 and the Staff for suggestions and support in revising the manuscript.
Best regards,
Ileana Vigentini